# In Quest of Molecular Materials for Quantum Cellular Automata: Exploration of the Double Exchange in the Two-Mode Vibronic Model of a Dimeric Mixed Valence Cell †

Boris Tsukerblat [1],*, Andrew Palii [2],* and Sergey Aldoshin [2]

1   Department of Chemistry, Ben-Gurion University of the Negev, Beer-Sheva 84105, Israel
2   Laboratory of Molecular Magnetic Nanomaterials, Institute of Problems of Chemical Physics of Russian Academy of Sciences, 142432 Chernogolovka, Moscow Region, Russia; sma@icp.ac.ru
*   Correspondence: tsuker@bgu.ac.il (B.T.); andrew.palii@uv.es (A.P.)
†   Dedicated to the memory of Professor Peter Day.

**Abstract:** In this article, we apply the two-mode vibronic model to the study of the dimeric molecular mixed-valence cell for quantum cellular automata. As such, we consider a multielectron mixed valence binuclear $d^2 - d^1$–type cluster, in which the double exchange, as well as the Heisenberg-Dirac-Van Vleck exchange interactions are operative, and also the local ("breathing") and intercenter vibrational modes are taken into account. The calculations of spin-vibronic energy spectra and the "cell-cell"-response function are carried out using quantum-mechanical two-mode vibronic approach based on the numerical solution of the dynamic vibronic problem. The obtained results demonstrate a possibility of combining the function of molecular QCA with that of spin switching in one electronic device and are expected to be useful from the point of view of the rational design of such multifunctional molecular electronic devices.

**Keywords:** quantum cellular automata; molecular cell; mixed-valence; electron transfer; double exchange; magnetic exchange; dimeric mixed valence clusters

## 1. Introduction

This article is dedicated to the memory of Professor Peter Day with the question posed to him in 1998 ([1], see image below): "Molecular information processing: Will it happen?" This question and subsequent discussions in his inimitable manner was focused on the fundamental issues of the "design and manufacture artificial structures using molecules that will carry out" the function of storing and processing memory in living organisms. In his general arguments, Peter Day appealed to common problems of information (see highlights in the excerpt from the article by Peter Day published in *Proc. Royal Inst. Great Britain*) interconnected with the switching processes in a binary systems and discussed the fundamental limits of computing speed and power dissipation. These ideas presented in detail along with the discussion of the molecular aspects are in focus of the contemporary issues in the topic of Quantum Cellular Automata (QCA) and search for the new molecular materials for the nanoscale devices. In this regard, it is pertinent to note that the Robin and Day assignment [2] of mixed-valence compounds according to the degree of localization plays a guiding role in the search of the relevant molecules.

In accordance with the general ideas proposed in the pioneering study by Lent et al. [3], the electronic QCA devices are based on the square planar cells composed of quantum dots [3–5]. Two excess electrons captured by a square-planar four-dot cell provide a possibility to encode binary information (**0** and **1**) in the two antipodal (diagonal) distributions of the charges. To illustrate encoding and operating with binary information underlying the actions of electronic devices, a dimeric system can be used, as illustrated in Figure 1. The dimeric unit can be considered as a "half-cell" from which the "full-cell" (tetrameric

unit) can be constructed. Figure 1 illustrates a dimeric cell in which the delocalized pair the mobile electron is evenly distributed between two sites and the two predominantly localized configurations corresponding to the binary **0** and **1**.

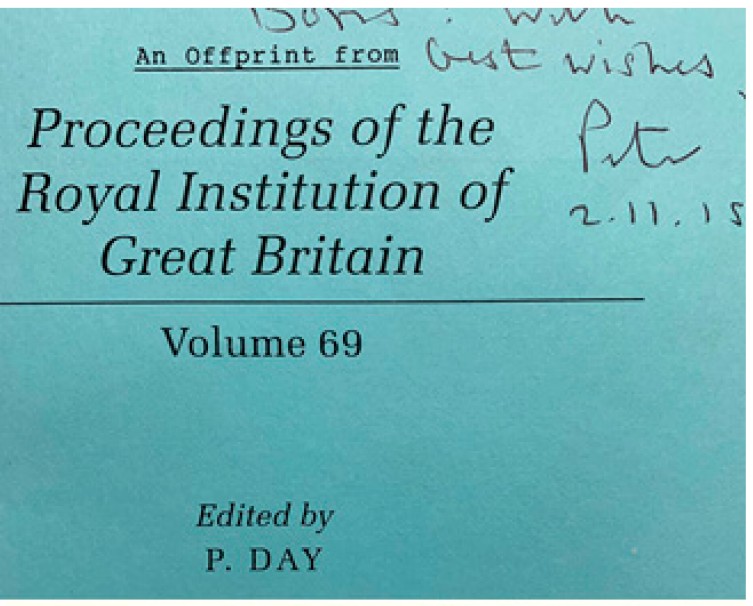

Excerpt from the article by Peter Day published in *Proc. Royal Inst. Great Britain*, v.69. pp. 85–106 (1998).

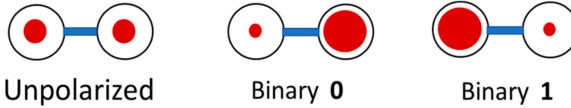

**Figure 1.** Two charge distributions in a two-dot cell or in a dimeric (mixed-valence) MV molecule with one mobile electron corresponding to the delocalized (unpolarized) configuration and localized configurations corresponding to binary and **1**. The red balls indicate the populated sites and their sizes symbolize the degree of localization of the mobile electrons.

The functional properties of devices are based on the concept of the action of the Coulomb forces that can control and transmit the binary information encoded in a cell. Let us consider the two dimeric cells 2 and 1 in a certain geometry (shown in Figure 2), one of which has a definite charge configuration (binary **1** in Figure 2), while the second one is unpolarized. Let us assume that the polarized state of the cell 2 can be induced and controlled so that this cell can be termed as the "driver cell". The electrostatic effect of the driver cell 2 with a given polarization affects the neighboring cell 1 forcing this cell to acquire polarization **0**. The polarization of the cell 1 obeys the action driver cell and in this way the driver cell can transmit the binary information to the surrounding cells. Thus the cell 1 can be referred to as the "working cell".

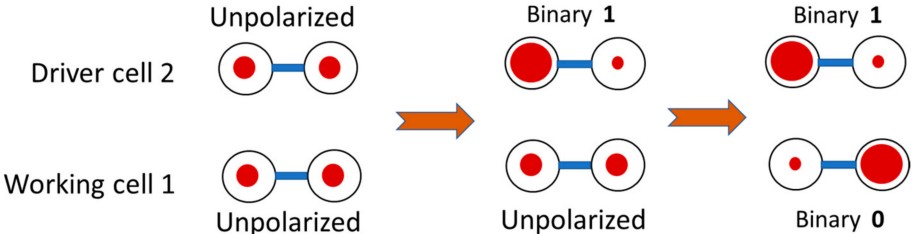

**Figure 2.** Scheme of the elementary process of the control of the binary information through the action of the dimeric driver cell 2 to the working cell 1. Left part shows the initial step of the information processing: unpolarized driver and unpolarized working cell. Then at the next step the driver gets polarization corresponding to the binary **1**. Finally, the polarized driver cell acts on the unpolarized working cell and gives rise to its polarization corresponding to binary **0**.

By combining such cells, one can obtain different QCA-based devices such as wires, majority logical gates, etc. QCA can be regarded as alternative to the traditional element base obtained with the aid of complementary metal–oxide–semiconductor (CMOS) technology for creating nanoscale devices capable of performing computations at very high switching rates. The advantages of the QCA devices as compared with CMOS ones are the smaller size of the devices and lower power consumption, which is a consequence of the current-less nature of the QCA devices.

As a further development of the concept of QCA based on quantum dots, a new fruitful idea of usage molecules as cells was proposed (see discussion in Reference [6] and a short overview in [7]). As natural candidates act as molecular cells, the dimeric MV molecules with one mobile excess electron and tetrameric MV molecules with two excess electrons have been proposed, as these bistable systems can encode the binary information [6]. The scope of this article does not allow to review a wealth of information on the results of molecular QCA, including the synthesis and study molecular systems, which are able to act as molecular cells. The reader can find it in references [8–25] and references thein.

In the systems considered until recently as tetrameric or dimeric cells (in both molecular and quantum dot-based implementations), the excess electrons were assumed to migrate over the diamagnetic centers (which can be alternatively referred to as "spinless cores"). The search for molecules that would be suitable for the design of molecular cells represents a problem whose complexity is caused by the requirements to such cells implied by their functional purposes, such as ability to encode the binary information and to easily

control it by varying external field of the driver-cell. In terms of physical concepts, this can be formulated as the requirement of clearly pronounced property of bistability of the charge distribution in the working cell, which would ensure binary information is encoded. Moreover the high polarizability allows operations to be performed with the encoded information in the working cell by means of variable electric field created by the driver-cell.

To-date, the reported molecules possessing all properties required for the design of QCA are relatively scarce given that the search and synthesis of suitable molecules represents a very non-trivial task. At the same, this task lays the core of the design of the molecular QCA. In this regard, recently we have proposed [26], in order to expand the class of systems suitable as cells by including MV clusters, by which the excess electrons move over the network of localized spins (spin cores).

The presence of spin cores in such kind of systems (which here will be conventionally referred to as magnetic clusters) leads to the appearance of a specific kind of magnetic interaction, known as double exchange (DE). The DE is a spin-polarization mechanism resulting in the ferromagnetic spin alignment that occurs in MV clusters, containing mobile excess electrons, which produce polarization of the localized spins hat, to explain the ferromagnetic properties of some perovskites (see classical paper [27]. As far as the magnetic ions are involved, the Heisenberg-Dirac-Van Vleck (HDVV) exchange interaction between these ions is considered as well. As distinguishable from the traditional cells in which only charges are employed, the magnetic clusters considered here the spin degrees of freedom can be involved. Therefore, along with the QCA function proper, an additional useful functionality can be expected, such as spin switching in the working cell under the action of the electrostatic field induced by the driver-cell. Actually, the magnetic working cell has a ground state with a definite full spin that can be changed under the action of the purely electrostatic field of the magnetic driver cell. This phenomenon has been referred to as "spin switching" effect.

In the recent short communication [26], only a general idea of using magnetic MV clusters as cells for QCA devices and spin switchers has been proposed and theoretically supported in the framework of a simplified model, which takes into account only relevant spin-spin interactions and electron delocalization. At the same time, a number of topical issues, related to the theory of cells in which the DE involved has not been discussed, is provided in reference [26]. In particular, the previously developed two-mode vibronic model of an one-electron dimeric cell [28] should be generalized to the case of magnetic dimeric cells exhibiting DE and HDVV exchange forms. The model takes into account the interaction of an excess electron with both "breathing" local vibrations and the intercenter vibration.

In this article, we consider this problem for the case of the magnetic dimeric cells of the $d^2 - d^1$–type based on the transition metal ions (system in which the electron transfer occurs over the paramagnetic spin cores $d^1$). Although, the results born much wider frameworks of applicability. The aim of the present study is to develop the vibronic model for a free magnetic cell and the cell influenced by the driver cell. We attempted to reveal the conditions for spin switching under the action of the Coulomb field induced in the working cell by a neighboring driver-cell. On the basis of the developed model, we discuss both the spin switching effect and its influence on the cell-cell response function.

## 2. Magnetic Interactions in a $d^2 - d^1$–type Cell

We consider a $d^2 - d^1$–type MV dimer A-B (Figure 3), containing two equivalent paramagnetic centers playing a role of spin cores and an excess electron, migrating between these cores. We denote the spin of the core ($d^1$ ion) by $S_0$ ($S_0 = 1/2$ in the present case). It is assumed that we are dealing with the high-spin metal ions so that the spin of $d^2$ ion is $S_0 + 1/2 = 1$. These two spins are combined to give the total spin $S$ of the dimer, which takes the values 1/2 and 3/2.

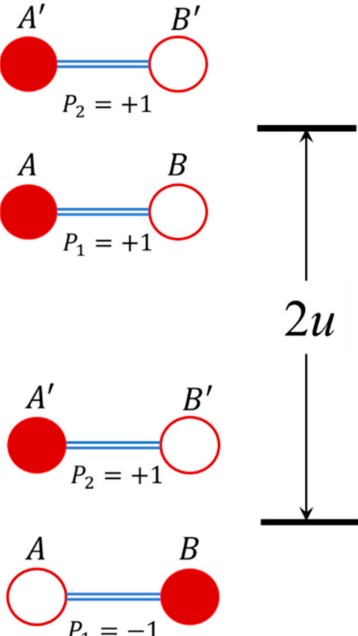

**Figure 3.** Mutual disposition of the driver-cell and the working cell, and the two possible electronic distributions in a pair of interacting dimeric cells shown to explain the physical meaning of the intercell Coulomb energy $u$. The sites belonging to the driver-cell and the working cell are primed and unprimed correspondingly, the site comprising (in a definite electronic distribution) the excess electron is shown as a red ball.

The electronic Hamiltonian of the dimer is represented as the following matrix defined for each value of the total spin:

$$\psi_A(S, M_S)\, \psi_B(S, M_S)\hat{H}(S) = \begin{pmatrix} -JS(S+1) + uP_2 & t_S \\ t_S & -JS(S+1) - uP_2 \end{pmatrix}. \quad (1)$$

In Equation (1) the basis $\psi_A(S, M_S)$ and $\psi_B(S, M_S)$ defining the $2 \times 2$-matrix includes the localized states (symbols $A$ and $B$), which are characterized by the total spin $S$ and quantum number of spin projection $M_S$. The off-diagonal matrix elements in Equation (1) are defined as follows:

$$t_S = \frac{t(S + 1/2)}{2S_0 + 1} \quad (2)$$

The value $t_S$ can be considered as the spin-dependent electron transfer parameter, or alternatively, DE parameter (deduced by Anderson and Hasegawa [27]), while $t$ is the one-electron (bi-orbital) transfer parameter. Linear spin dependence of the transfer matrix element in Equation (2) is known as the main manifestation of the DE, and just this linear dependence predetermines ferromagnetic effect caused by this interaction. Diagonal matrix elements, include two types of contributions, namely, the exchange contribution of HDVV type, where $J$ is the exchange parameter describing the interaction of centers with spins $S_0$ and $S_0 + 1/2$ (within each localized configuration of the dimer). In order to take into account the effect of the driver cell, we consider also the contribution describing the Coulomb interaction between the cells (terms (terms $\pm uP_2$), where $P_2$ is the polarization of the driver- cell A$'$ $-$ B$'$ (cell 2). The value $P_2 = +1$ corresponds to localization of the mobile charge on site A, while providing $P_2 = -1$ the charges localized on site B, as shown in Figure 3. Usually, in the theory of QCA, it is assumed that the driver-cell is a source of a Coulomb field acting on the working cell (cell 1), which causes its polarization. Finally, $u$ is the characteristic energy of the Coulomb interaction between the cells. The physical meaning of this parameter is clear from Figure 3 showing the relative disposition of the working cell and the driver-cell, and the two possible electronic distributions in a pair

of interacting cells. As observable, the energy $2u$ is the difference between the Coulomb repulsion energies of the excess electrons occupying neighboring and distant (energetically favorable) positions in the two interacting dimers.

### 3. Polarization of a Cell

The polarization of the driver-cell is determined by the following expression [3,4]:

$$P_2 = \frac{\rho_{A'} - \rho_{B'}}{\rho_{A'} + \rho_{B'}} \tag{3}$$

where $\rho_{A'}$ and $\rho_{B'}$ are the probabilities (electronic densities) of the two localizations of the excess electron ($\rho_{A'} + \rho_{B'} = 1$). The electronic densities on the centers have standard quantum-chemical definition through the eigenvectors of the system. It is assumed that polarization of the driver cell (in conformity with the definition of the driver cell) can be varied in a controllable manner from the value $P_2 = -1$ to value $P_2 = +1$, i.e., between the two fully polarized states. When the driver cell 2 is polarized, it induces polarization of the working cell 1. The latter polarization is also determined by Equation (3) in which the replacement $\rho_{A'} \rightarrow \rho_A$, $\rho_{B'} \rightarrow \rho_B$ is to be made.

While, the DE in dimers are known to always involve a ferromagnetic interaction [27], the HDVV exchange can, either be ferro- or antiferro-magnetic, depending on the physical conditions which determine the sign of the parameter $J$. We consider the most common situation when the HDVV exchange is antiferromagnetic ($J < 0$). In this case, the ground spin-state of the free cell (case of $P_2 = 0$) is determined by the competition between the ferromagnetic DE and antiferromagnetic HDVV exchange interactions. In most cases, DE dominates over the HDVV exchange and so the ground state proves to be ferromagnetic ($S_{gr} = 2S_0 + 1/2 = 3/2$). When $P_2 \neq 0$ the working cell is subjected to the action of an electrostatic field created by the driver-cell, which tends to localize the excess electron in the working cell. Since this field restricts the mobility of the excess electron it leads to a partial suppression of the ferromagnetic DE. As a result of such suppression, the antiferromagnetic HDVV exchange (that acts within localized configurations and hence is not affected by the field) can become the dominant interaction, which can lead to a stabilization of the spin-state with lower total spin value or, in other words, to cause a spin switching effect.

Effective control of the spin-state of the working cell by the Coulomb field of the driver-cell can be significantly complicated by the fact that, in strongly delocalized MV systems the ground ferromagnetic state is separated from the excited state, with a lower spin by a quite large energy gap and so spin switching is only possible is the Coulomb field is so strong that its effective energy exceeds this gap. However, the last condition is difficult to fulfill since the distances between the neighboring cells must be much longer than the distances between centers inside the cell, in order to prevent the electron transfer and the HDVV exchange between the cells. At the same time, the situation may be not so hopeless if we are dealing with the MV molecular systems, in which along with the electronic interactions, an essential role is played by the interaction of excess electrons with molecular vibrations of the cell. Therefore, we arrive at the conclusion regarding the crucial role of the vibronic coupling in MV molecules, in the context of discussion of the feasibility of spin switching effect.

### 4. Two-Mode Vibronic Model

In the vibronic model of a MV unit, we employ the interactions of the excess electron with the two types of vibrations that are taken into account. The first type is the active molecular vibration, which is composed of the totally symmetric "breathing" local modes spanning non-magnetic atoms of the redox sites. In inorganic metal clusters, the local vibrations are related to the displacements of the nearest ligand environments of the metal ions, while in organic compounds the redox sites have more complex structure and can involve several C-C bonds. In a particular case of the octahedrally coordinated metal sites in inorganic compounds, the local modes can be assigned to the full-symmetric vibrations ($A_{1g}$

symmetry). The vibronic interaction in MV compounds (particularly, in molecular cells) can be described in the framework of generally accepted Piepho-Kraus-Schatz (PKS) vibronic model [28–30]. Although this model is rather simplified, it successfully describes the key features of MV systems, such as the occurrence of a potential barrier between localized configurations. In particular, this model underlies the Robin and Day classification of MV compounds according to the degree of localization.

Within the conventional PKS vibronic model, the following symmetric and antisymmetric (with respect to inversion in the dimer resulting in the interchange $A \leftrightarrow B$) molecular coordinates can be composed of the local dimensionless coordinates $q_A$ and $q_B$ des cribing the "breathing" displacements [29,30]:

$$q_{\pm} = \frac{1}{\sqrt{2}}(q_A \pm q_B) \tag{4}$$

It can be shown that the totally symmetric (even) vibration $q_+$ can be excluded from the consideration since the corresponding contribution to the vibronic coupling is proportional to the unit matrix. From the physical point of view, this means that in course of this vibration, both sites are compressing and expanding in phase. This is obviously irrelevant to the charge transfer processes. On the contrary, the antisymmetric coordinate is interconnected with the vibration in course of which the expansion and compression of the sites occurs in the out-of-phase manner. The expanded site traps the mobile charge, while the compressed site tends to push it out. and thus, the PKS coupling is closely related to the electron transfer processes. The frequency of the vibration $q_-$ will be denoted by $\omega$ and the parameter of vibronic coupling with this mode by $v$.

The second type of molecular vibrations involved in the model we use here is interrelated with the change of mutual disposition of the redox sites without changing their sizes. The corresponding part of the vibronic coupling describes the interaction of the excess electron with an intercenter vibration, which changes the distance between the sites [30]. Such kind of coupling arises from the modulation of the transfer parameter caused by the change of distance between the redox sites. To deduce this part of the vibronic coupling one should represent the transfer integral as a following series expansion:

$$t(R) = t(R_0) - \zeta (R - R_0) + \cdots \tag{5}$$

where $\zeta = -(\partial t / \partial R)_{R=R_0}$ is the parameter of vibronic interaction, and $t(R_0) \equiv t$ is the value of the transfer parameter evaluated at the equilibrium distance $R_0$ between the ions (below we will simply term it "transfer parameter"). The quantity $R - R_0$ plays a role of the vibrational coordinate associated with the inter-center vibration. We denote the frequency of this vibration as $\Omega$ and introduce the corresponding dimensionless normal coordinate as $Q = (R - R_0)/\sqrt{\hbar/M\Omega^2}$, where $M$ is the effective mass. In contrast to local vibrations, which change the sizes of interacting mononuclear fragments, $A$ and $B$ as assumed in the PKS model, the inter-center vibration leaves the sizes of the coordination spheres unchanged. Therefore, we arrive at the two-mode vibronic problem, which takes into account the totally symmetric vibration $Q$ with a frequency $\Omega$ and the antisymmetric (odd) vibration $q_-$ with the frequency $\omega$.

Then, the total Hamiltonian of a dimeric working cell subjected to action of the Coulomb field created by the adjacent driver-cell with polarization $P_2$ is obtained in the following matrix form (see Refs. [26,28]):

$$\hat{H} = \left[ \frac{\hbar\omega}{2}\left(q^2 - \frac{\partial^2}{\partial q^2}\right) + \frac{\hbar\Omega}{2}\left(Q^2 - \frac{\partial^2}{\partial Q^2}\right) \right] \begin{pmatrix} 1 & 0 \\ 0 & 1 \end{pmatrix} + \begin{pmatrix} vq + uP_2 & t_S - \zeta_S Q \\ t_S - \zeta_S Q & -vq - uP_2 \end{pmatrix}. \tag{6}$$

In Equation (6) the following short notation is used:

$$\zeta_S = \frac{\zeta\left(S + \frac{1}{2}\right)}{2S_0 + 1} \tag{7}$$

The value $\zeta_S$ can be referred to as the spin-dependent coupling parameter with the intercenter vibration, which has the same spin dependence as the DE contribution in Equation (2). The matrices involved in Equation (6) are defined in the same bi-dimensional basis $\psi_A(S, M_S)$, $\psi_B(S, M_S)$ as the matrix of the electronic Hamiltonian, Equation (1). Equation (7) represents a block of the full Hamiltonian matrix with a definite set of spin quantum numbers $S, M_S$.

## 5. Dynamic Vibronic Problem

The commonly accepted tool in considering the energy pattern and electron localization in MV cluster is the adiabatic approximation, based on the assumption that the kinetic energy of the heavy ions can be neglected, or alternatively, that the nuclear motion is much slower than the electronic one. In this approach, the energy levels of the system are associated with the adiabatic potentials or potential curves. The applicability of the adiabatic approximation is invalid for the vibronic levels in the vicinity of the avoided crossing of the potential curves. This area is relevant to the process occurring when the localization in the working cell changes under the action of the driver cell. That is the reason why the subsequent analysis is based on the solution of the dynamic vibronic problem for full electron-vibrational Hamiltonian including kinetic energy of the ions. The importance of the non-adiabatic approach in the problem of mixed valency (for a free MV dimer) was realized long time ago (see reference [29] dealing with the quantum-mechanical evaluation of the profiles of the intervalence absorption) and applied to the study of a molecular cell for QCA in reference [28].

To solve the quantum-mechanical problem the matrix of the Hamiltonian, Equation (6), is to be presented in the basis composed of the products $\psi_A(SM)|nN\rangle$ and $\psi_B(SM_S)|nN\rangle$, where $|nN\rangle$ are the wave functions of the two-dimensional harmonic oscillator (first term in Equation (6)), $n$ and $N$ are vibrational quantum numbers related to the two types (PKS and inter-center) of vibrational modes under consideration. These functions are the eigen-functions of the unperturbed Hamiltonian that is the Hamiltonian from which the vibronic coupling is eliminated. To obtain a solution to the dynamic vibronic problem, diagonalization of this infinite matrix is required. The numerical solution in the truncated basis gives a set of spin-vibronic energy levels $\varepsilon_k^s$ of the working cell and the corresponding spin-vibronic wave functions, which have the form of the following superpositions:

$$|k, S, M_S\rangle = \sum_{n,N} \left[ c_{A,n,N}^{k,\,S} \, \psi_A(S, M_S)|n, N\rangle + c_{B,n,N}^{k,S} \, \psi_B(S, M_S)|n, N\rangle \right]. \tag{8}$$

The truncation procedure restricts the size of the matrices to be diagonalized in such a way that it ensures a required accuracy (i.e., good convergence) in the evaluation of the low lying vibronic levels. Knowledge of the coefficients in the eigen-functions in Equation (8) means we can assess the electronic densities on the sites that are required for the evaluation of the polarization of the cell and consequently the cell-cell response function.

The dependences of the spin-vibronic wave functions on the polarization $P_2$ of the driver cell are then calculated for various sets of parameters and used to evaluate the key characteristics of the QCA cell (and QCA gate), known as the cell-cell response function, that is the dependence of the polarization $P_1$ of the working cell on the polarization $P_2$ of the driver-cell.

## 6. Spin-Vbronic Levels and Cell-Cell Response Function

Figure 4 shows the dependences of the spin-vibronic energy levels of the working cell represented by a MV dimer of $d^2 - d^1$–type on the polarization $P_2$ of the driver-cell calculated for fixed values of the parameters $t$, $J$ and $u$ and various ratios between the values of the vibronic parameters $\zeta$ and $v$. Figure 5 shows a family of the cell-cell response functions calculated with the same sets of parameters. To simplify the discussion the frequencies of the two modes are assumed to be equal ($\Omega = \omega$).

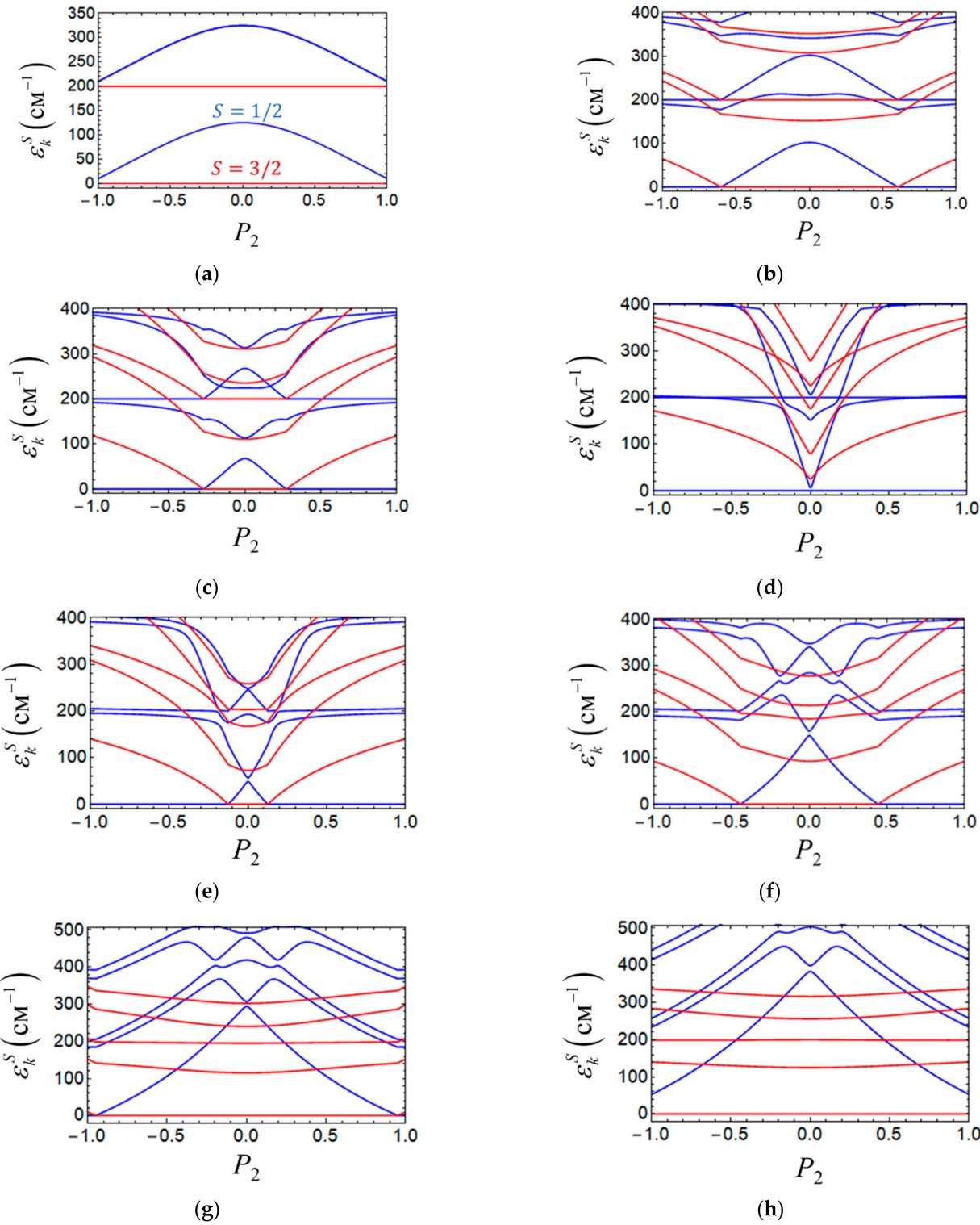

**Figure 4.** Spin-vibronic energy levels of the working cell $d^2 - d^1$ calculated as a function of polarization of the driver cell $P_2$ at $u = 600$ cm$^{-1}$, $\hbar\omega = \hbar\Omega = 200$ cm$^{-1}$, $t = 1000$ cm$^{-1}$, $J = -125$ cm$^{-1}$ and the following sets of the vibronic coupling parameters: $v = 0$, $\zeta = 0$ (**a**); $v = 300$ cm$^{-1}$, $\zeta = 0$ (**b**); $v = 400$ cm$^{-1}$, $\zeta = 0$ (**c**); $v = 500$ cm$^{-1}$, $\zeta = 0$ (**d**); $v = 500$ cm$^{-1}$, $\zeta = 200$ cm$^{-1}$ (**e**); $v = 500$ cm$^{-1}$, $\zeta = 300$ cm$^{-1}$ (**f**); $v = 500$ cm$^{-1}$, $\zeta = 400$ cm$^{-1}$ (**g**); $v = 500$ cm$^{-1}$, $\zeta = 450$ cm$^{-1}$ (**h**). The ground spin-vibronic level is chosen as a reference point for the energy.

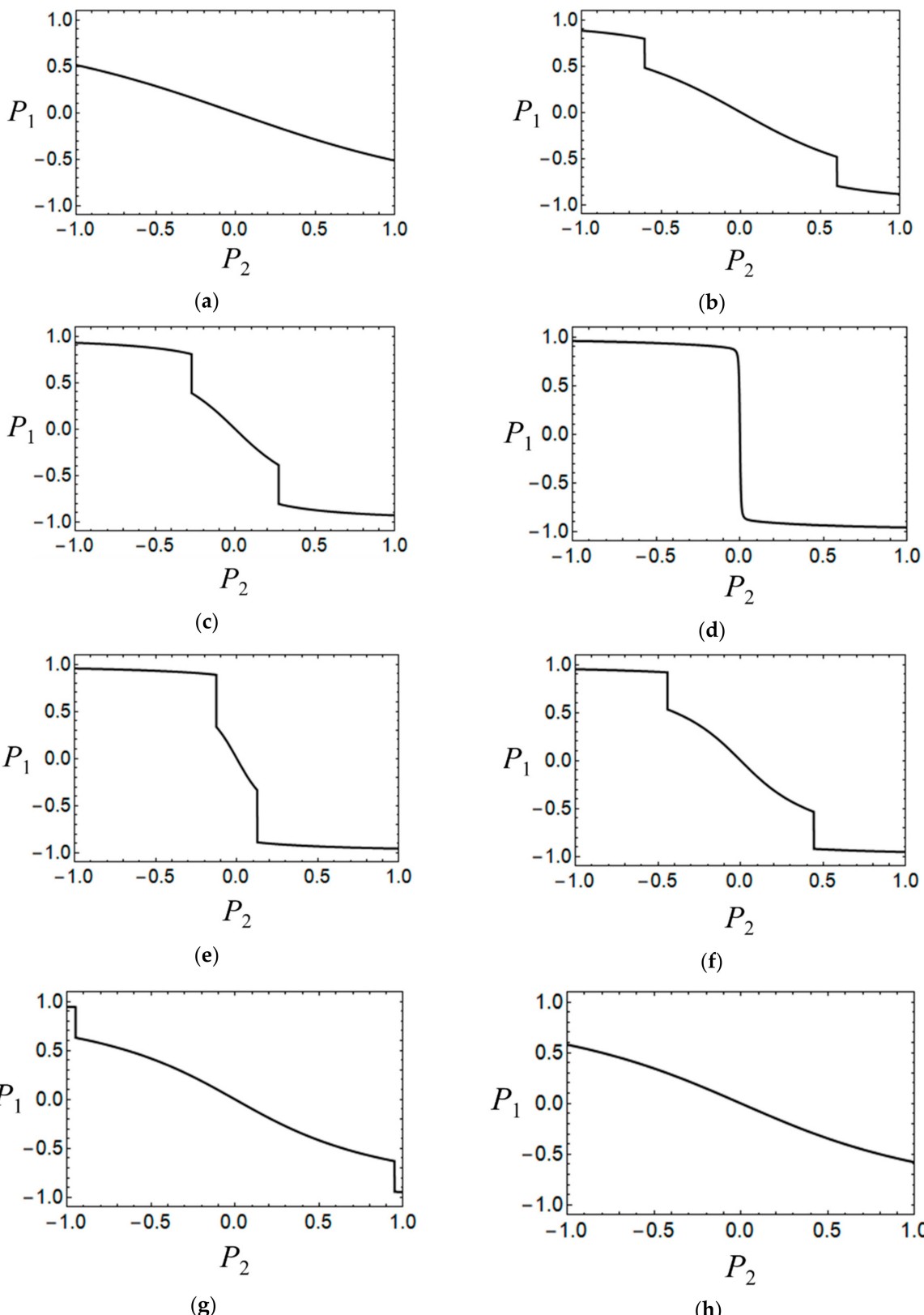

**Figure 5.** Cell-cell response functions evaluated for the $d^2 - d^1$–type cells at $u = 600\ \text{cm}^{-1}$, $\hbar\omega = \hbar\Omega = 200\ \text{cm}^{-1}$, $t = 1000\ \text{cm}^{-1}$, $J = -125\ \text{cm}^{-1}$ and following sets of the vibronic coupling parameters: $v = 0$, $\zeta = 0$ (**a**); $v = 300\ \text{cm}^{-1}$, $\zeta = 0$ (**b**); $v = 400\ \text{cm}^{-1}$, $\zeta = 0$ (**c**); $v = 500\ \text{cm}^{-1}$, $\zeta = 0$ (**d**); $v = 500\ \text{cm}^{-1}$, $\zeta = 200\ \text{cm}^{-1}$(**e**); $v = 500\ \text{cm}^{-1}$, $\zeta = 300\ \text{cm}^{-1}$(**f**); $v = 500\ \text{cm}^{-1}$, $\zeta = 400\ \text{cm}^{-1}$(**g**) and $v = 500\ \text{cm}^{-1}$, $\zeta = 450\ \text{cm}^{-1}$(**h**).

Figure 4a shows the limiting case of a negligibly weak coupling with both types of the vibrational modes ($v = \zeta = 0$). The selected sample values $t = 1000 \, \text{cm}^{-1}$, $J = -125 \, \text{cm}^{-1}$ of the electronic parameters correspond to the indicated above-typical situation when the ferromagnetic contribution of DE significantly exceeds the antiferromagnetic contribution of the HDVV exchange. In this case, the ground state of an isolated cell $P_2 = 0$ has a maximum spin value $S = 3/2$ and it is separated from the first excited state with $S = 1/2$ by the energy gap of around 130 cm$^{-1}$. Figure 4a shows that the Coulomb field of the driver-cell tends to decrease the gap between these states due to suppression of the DE, but the ground state, in this case, remains at $S = 3/2$, even at the maximum polarization ($|P_2| = 1$) of the driver-cell.

The energy spectra in Figure 4b–d illustrate the influence of the PKS-type vibronic interaction on the field dependences of the energy levels. Since the PKS-type vibronic interaction tends to localize the excess electron, it weakens the ferromagnetic effect of the DE so that at a certain value of $|P_2|$ the spin switching $S = 3/2 \rightarrow S = 1/2$ occurs. It is notable that the larger the value of $v$, the lower the value of $|P_2|$, by which such spin switching occurs. This can be seen from the comparison of Figure 4b,c. Finally, for a sufficiently strong PKS-type vibronic coupling (the case shown in Figure 4d) the DE turns out to be almost completely suppressed and the ground spin-vibronic state is the low-spin one even provided that $P_2 = 0$.

Figure 4e–h demonstrate the effect of coupling of the excess electron with the inter-center vibration while the coupling with the PKS mode is assumed to be strong (the same as in the case shown in Figure 4d). As it follows from Figure 4e–h, the interaction with the inter-center vibration produces an effect that is in some sense opposite to the effect of the PKS interaction. Indeed, the PKS-type vibronic coupling tends to localize the electron at one of the redox sites, while the vibronic coupling with the intercenter mode promotes delocalization. In turn, an increase in the degree of delocalization leads to the enhancement of the ferromagnetic effect of the DE. As a result, the high-spin ground state of the working cell is restored, and at the same time the conditions required for the manifestation of spin switching become valid. Consequently, with an increase of the parameter $\zeta$, the critical field (i.e., the value $|P_2|$) at which spin switching occurs also increases, which can be clearly seen through the comparison of the energy patterns, as shown in Figure 4e,f,g. Finally, in the case of a sufficiently strong interaction with the intercenter vibration, the self-trapping effect of the PKS-interaction proves to be fully compensated (Figure 4h), and consequently, the ground state remains the high-spin one regardless of the magnitude of $|P_2|$ exactly as in the case of a vanishingly weak vibronic coupling shown in Figure 4a.

The above effects of the two types of vibrations on the spin-vibronic energy levels of the cell manifest themselves also in the shapes of the cell-cell response function $P_1(P_2)$ (Figure 5). First, from Figure 5a,h, in the case of negligibly, we see weak vibronic PKS coupling. In the case of strong coupling with the inter-center vibration, the cell exhibits weak monotonic and almost linear response to the Coulomb field of the driver-cell. This feature of the cell-cell response function indicates that the cases mentioned so far are the least favorable ones from the point of view of functioning of the system, as both QCA cell and spin switcher.

In contrast, in the case of a strong PKS interaction and vanishingly weak interaction with the inter-center mode (case shown in Figure 5d), the cell-cell response function demonstrates a sharp nonlinear behavior, even at a very weak change in the polarization of the driver cell. This strongly non-linear behavior is indicative of the case for strong localization, which occurs when PKS coupling is strong. This case is the most favorable one for the functioning of the QCA devices.

Finally, in the intermediate cases (Figure 5b,c,e–g) the response function behaves non-monotonically demonstrating an abrupt change at critical values $|P_2|$ at which the spin switching in the cell occurs. This behavior is due to the fact that the states with lower values of the total spin are characterized by a higher polarizability as compared to the

states for which the spin is higher. These cases are of practical interest from the point of view of the prospects in designing spin switching devices based on magnetic MV dimers.

## 7. Conclusions

This study was devoted to the problem of molecular implementation of QCA, a perspective technology with promising applications. Here, we attempted to proceed in studying the molecular cells as the central ingredient in the design of the QCA logical gates. We consider a cell represented by the multi-electron mixed valence binuclear $d^2 - d^1$–type cluster in which the double exchange, as well as the Heisenberg-Dirac-Van Vleck exchange interactions are operative. The dimeric unit can be considered also as a part of the bi-dimeric cell encoding binary information in the antipodal charge distributions.

Since the information is encoded in the charge distribution, an important issue we studied is interrelated with the vibronic coupling is known as a factor determining the charge localization in MV systems. We propose the two-mode vibronic model involving interactions with the local and intercenter modes and involves also the DE and HDVV exchange interactions have been applied to analyze the functional properties of MV dimer as the QCA cell.

It is demonstrated that the magnetic cell can encode the binary information and formulated the favorable conditions under which this functionality is efficient. An essentially new functionality closely interrelated with the spin degrees of freedom is the feature of spin switching in the working cell that was shown to occur under the electrostatic field of the driver cell. Therefore, it was shown that the magnetic cell can exhibit the property of multi-functionality being a reservoir for the binary information, and at the same time, act as a spin switcher.

The influence of both kind of vibrations on the dependences of the spin-vibronic levels of the working cell on the polarization of the driver-cell has been studied. The local "breathing" vibrations produce the trapping effect and increase the non-linearity of the cell-cell response function, while the inter-center vibrations tend to delocalize the system and therefore have destructive influence for the cell-response. Based on the developed vibronic model, the new features of the cell-cell response functions interrelated with the DE and HDVV exchange interactions in the magnetic molecular cell based on binuclear MV clusters of the $d^2 - d^1$–type have been revealed:

(1) in the case of the dominating DE, the character of the ground spin-state of a free and polarized cells has been shown to be strongly influenced by the interactions of the electronic subsystem with both types of vibrational modes included in the model. Therefore, in the case of vanishingly weak vibronic interaction, as well as in the case of strong coupling with the intercenter vibration, the ferromagnetic effect produced by the DE proves to be the dominating interaction;

(2) in this case of strong DE the $P_1(P_2)$ dependence (cell-cell response function) demonstrates weak and almost linear cell-cell response that is an unfavorable case for the QCA function. Moreover, the spin-switching is not possible in this case;

(3) in contrast, at strong vibronic PKS coupling, the ferromagnetic DE is largely suppressed, which leads to the stabilization of the state with a minimum total spin, along with the appearance of a strong nonlinear cell-cell response. This case is definitely favorable for the design of QCA-based devices.

(4) finally, when the contributions of both types of vibrations are comparable, the magnetic cell has been shown to exhibit properties of a spin-switcher, i.e., the electrostatic field of the driver change spin of the working cell along with the charge distribution. In relation to the QCA application, this feature manifests itself in the specific shape of the cell-cell response function exhibiting sharp steps. More precisely, the polarizability of the working cell is efficiently increased that is favorable for the QCA action.

The results, mentioned so far, create hope on their practical feasibility and relevance to the rational design of multifunctional molecular electronic devices that combine the function of the charge carriers of information in QCA with that of the spin-switchers.

**Author Contributions:** Conceptualization, B.T., A.P. and S.A.; methodology, A.P.; software, A.P.; writing—review and editing, B.T. and A.P.; All authors have read and agreed to the published version of the manuscript.

**Funding:** This research was funded by Ministry of Science and Higher Education of the Russian Federation, grant number AAAA-A19-119092390079-8.

**Institutional Review Board Statement:** Not applicable.

**Informed Consent Statement:** Not applicable.

**Data Availability Statement:** Not applicable.

**Acknowledgments:** A.P. and S.A. acknowledge support from the Ministry of Science and Higher Education of the Russian Federation (the state assignment no. AAAA-A19-119092390079-8).

**Conflicts of Interest:** The authors declare no conflict of interest.

## Abbreviations

| | |
|---|---|
| QCA | Quantum Cellular Automata |
| CMOS | Complimentary Metal-Oxide-Semiconductor Structure |
| MV | Mixed Valence |
| DE | Double Exchange |
| HDVV | Heisenberg-Dirac-Van Vleck (exchange, model) |
| PKS | Piepho-Kraus-Schatz (coupling, model) |

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
