# Peer review of "In Quest of Molecular Materials for Quantum Cellular Automata: Exploration of the Double Exchange in the Two-Mode Vibronic Model of a Dimeric Mixed Valence Cell†"

_magnetochemistry, doi:10.3390/magnetochemistry7050066_

Round 1

Reviewer 1 Report

The motivation for the article is admirably written.  The authors are outstanding specialists in quantum chemical calculations. 

My comments are related to the style of description of their results and their interpretation.

  1. Unfortunately, the paper does not contain any references to experimental works in which the model of such dimer type is used to describe the experimental data. Judging by values of the parameters used in calculations, the information on characteristics of the dimer Cr2+-Cr3+ obtained in papers [1-3]  would be useful in this connection.
  2. The difference between the terms working cell and driver-cell is not clear. What electronic configurations of the dimer do they correspond to? Which dimer electronic states correspond to symbols A' - B' and how do they differ from A - B ?   The authors use the term spin switching .  What precisely is the meaning of the term?  What physical characteristic (parameter) of the dimer changes and under what external (internal) influence?
  3. The conclusions of the article are written carelessly and not specifically.

In summary, the text, Fig.1, conclusions need clarification.

Author Response

We have carefully taken into account your comments and suggestions. Your comments have been highly useful and allowed us to improve the manuscript. We thank the reviewers. Please find the attachment for checking.

Reviewer 2 Report

This paper discusses a new way of quantum control, using a Coulomb field to control spin switch in dimeric cells of d2 - d1 type. The authors give an overview of the magnetic interactions in a d2-d1 cell, the polarisation of the cell and describe the two-mode model. They then proceed to the dynamics and study the behaviour of the system under a combination of control variables; most noteworthy they demonstrate that in the case of strong PKS interaction and weak interaction with the intercepter mode, the system behaves as a QCA.

Overall, the presentation is well structured and the methodology well laid out supporting the results. However, there are various points where the paper needs further support or references, which are quite few.

Key points:

  -Equation 3: This may be a basic definition, but it would nice to have a reference here, especially for the non-expert reader.

  -Equation 6: It needs editing so that it appears centred in the text. Moreover, a bibliography reference would be good to be here as well.

 -Equation 8 and all the methodology described in the previous paragraph should be appropriately referenced

Section 4. “Breathing” parameters are referenced, but a short explanation about what they entail should also be given. Also the second paragraph needs referencing.

Generally, there are not a lot of references; the vast majority of researchers working on quantum control are familiar with concepts of solid state physics, electronics etc. This paper introduces novel concepts for quantum control using mechanisms commonly encountered in the domain of quantum chemistry which can make it hard to follow for many non-experts.

Moreover, recent articles on various QCA architectures and implementations, different to this one, should be referenced in the introduction to present a wider picture of the recent developments on the field.

Some further proofreading is also needed (example: in Section 6 pg 9 it reads: “Just this strongly non-linear behavior is indicative........... localization, which occurs prodiced that PKS coupling is strong.”

Author Response

(The authors gave the same response as above.)
